# High-Temporal-Resolution Corrosion Monitoring in Fluctuating-Temperature Environments with an Improved Electrical Resistance Sensor

**DOI:** 10.3390/s25010268

**Published:** 2025-01-06

**Authors:** Mao Takeyama

**Affiliations:** Meteorology and Fluid Science Division, Central Research Institute of Electric Power Industry, 1646 Abiko, Abiko-shi 270-1194, Chiba, Japan; takeyama3826@criepi.denken.or.jp; Tel.: +81-70-6576-0603

**Keywords:** electrical resistance method, atmospheric corrosion sensor, outdoor monitoring, corrosion depth, corrosion rate

## Abstract

The electrical resistance (ER) method is widely used for atmospheric corrosion measurements and can be used to measure the corrosion rate accurately. However, severe errors occur in environments with temperature fluctuations, such as areas exposed to solar radiation, preventing accurate temporal corrosion rate measurement. To decrease the error, we developed an improved sensor composed of a reference metal film and an overlaid sensor metal film to cancel temperature differences between them. The improved sensor was compared with an existing sensor product in outdoor monitoring experiments. The spike-like error during the daytime was successfully reduced. Furthermore, by utilizing a data-filtering process, we measured the corrosion rate every hour. Hourly corrosion rate measurements were difficult when the average daily corrosion rate was less than 50 µm/year under conditions of 0.05 g/m^2^ salt. Observations showed a strong correlation between corrosion rate and sensor surface humidity. In the future, this method will make it possible to study the relationship between the atmospheric corrosion rate and environmental changes over time.

## 1. Introduction

In this study, an improved version of the electrical resistance (ER) sensor, which can acquire corrosion rates even under temperature fluctuations by eliminating the temperature difference between the reference part and the sensor part, was developed. Until now, corrosion rates could not be acquired with a high temporal resolution in environments with temperature fluctuations, e.g., outdoors, because of errors caused by temperature differences.

There are various methods for measuring atmospheric corrosion rates [1,2], but the standard method is the metal coupon mass loss procedure, which is used for the assessment of outdoor corrosivity [3]. However, in the method of using metal coupons, the corrosion is evaluated by measuring the weight lost between installation and retrieval, so the obtained mass loss is only an integrated value over that period. Consequently, the corrosion rate during that period is only an average value. Even if the number of coupons installed is increased and measurement of mass loss is carried out multiple times, this method still has a low time resolution.

Real-time monitoring methods include galvanic corrosion sensing [4,5,6], electrochemical impedance spectroscopy (EIS) [7,8], and electrical resistance sensing. Electrochemical measurements such as galvanic sensing and EIS produce values that indicate the corrosion rate at a given moment. ER sensors, on the other hand, continuously measure the amount of corrosion loss and determine the corrosion rate from its derivative. The advantages of ER sensors include the simplicity of the measurement principle and operation and the fact that, unlike electrochemical measurements, they do not require the presence of electrolyte on the measurement surface, making them suitable for long-term continuous monitoring. ER sensors are also characterized by their flexible shape, capacity for in situ and continuous measurement, and ability to directly measure the corrosion depth. Thanks to those advantages, these sensors are widely used for outdoor monitoring [9,10] and fundamental corrosion experiments [11,12]. The sensor shape is not limited to a track film on a substrate [13] but can take various forms, such as a cable [14] or a pipe ring [15].

The atmospheric corrosion rate measured by the ER method has been confirmed to be almost the same as that by the coupon weight loss method, and a comparison test with the electrochemical impedance method showed that the ER method is more suitable for monitoring in low-humidity environments where the liquid film is very thin [9].

The sensitivity of the measurement of the corrosion depth and the durability of the sensors can be adjusted by changing the thickness of the metal film used in the sensors. For example, in cultural heritage institutions, very small corrosion depths of several nm can be measured [16], and this is a sufficiently accurate method that has been commercialized. Li et al. designed an ER sensor with multiple-line metal films to detect localized corrosion in steel [17].

As the electrical resistance of metals is strongly influenced by temperature, the ER sensor has a sensor part and a reference part to cancel the effects of this temperature fluctuation. The sensor and the reference part have the same shape and are installed next to each other, but the reference part is covered to prevent corrosion. Under the assumption that both parts are at the same temperature, the ratio of the electrical resistances is calculated, and the corrosion depth is calculated from the change in the ratio from the initial state using the following formula:(1)$$\Delta h=h_{ref,init}\left(\frac{R_{ref,init}}{R_{sens,init}}-\frac{R_{ref}}{R_{sens}}\right),$$
where Δ*h* is the corrosion depth of the metallic sensor; $h_{ref,init}$ is the initial thickness of the sensor part; $R_{sens}$ and $R_{ref}$ are the resistances of the sensor and reference parts at the time of measurement, respectively; and $R_{sens,init}$ and $R_{ref,init}$ are the initial resistances of the sensor and reference parts, respectively. To cancel the effects of temperature variations, the ratio of the resistances of the sensor and the reference parts is calculated.

Here, errors occur because of the difference in temperature between the sensor part and the reference part in outdoor and other environments where temperature fluctuations are large, making it impossible to measure the corrosion depth accurately with a high temporal resolution.

For example, it was difficult to measure short-term corrosion rates because of severe errors in measurements taken outdoors [18] (Figure 4) or in environments with temperature fluctuations inside constant-temperature and -humidity chambers [19] (Figure 3). Monitoring the atmospheric corrosion using an ER sensor, in a study by Li et al., allowed for the measurement of noise with a 24 h cycle, and data processing was used to remove the noise, allowing for the calculation of the corrosion rate [10].

This is because the sensor and the reference parts are located next to each other but in different positions, and also because the coating provided prevented the reference part from corroding. This means that RCM sensors have only been able to measure corrosion rates under conditions of a constant temperature for several hours or more. However, in an outdoor environment, the temperature changes cyclically on a daily basis, and there are also temperature spikes due to solar radiation. Under these conditions, long-term corrosion rates can be obtained, but it is impossible to measure corrosion rates that change occasionally.

In this study, the positional relationship between the two parts was designed to eliminate temperature differences to the greatest possible extent and reduce errors. Furthermore, measurements were carried out outdoors with the improved sensor and the existing product to compare the two and to show the potential of the improved sensor for outdoor use. In particular, the challenge was to carry out measurements of the corrosion rate every hour.

## 2. Materials and Methods

### 2.1. Improved ER Sensor

The design of the sensor is shown in Figure 1. The sensor is composed of, from top to bottom, the sensor part, an adhesive, the reference part, the adhesive, and a substrate. Because the sensor part and the reference part overlap, instead of being located next to each other as in previous designs, the temperature differences between the two parts can be eliminated by thermal conduction. This design reduces errors caused by temperature differences and also provides a covering for the reference part. The sensor part and the reference part cannot be in direct contact, but to ensure the best possible heat transfer, the adhesive should be as thin as possible with electrical insulation. In this study, double-sided tape (150 µm thickness, 3 M) was used, which consisted of a silicone-based adhesive on both sides of a polyimide tape.

The metal used was 50 µm thick iron foil of the shape shown in Figure 1. Acrylic with a thickness of 10 mm was used as the substrate.

### 2.2. Experimental Setup

Comparative tests were carried out using product ER sensors (Schlinks, Japan). The product sensor model had a thickness of 200 µm, a 2 mm thick iron track, and a sensor size of approximately 45 mm, which is smaller than that of the improved sensor.

As shown in Figure 2, the two sensors are designed differently. In the existing product in the right panel, a reference part of the same shape is installed under a white cover next to the exposed sensor part. Meanwhile, in the improved sensor on the left, the same-shaped reference part is present under the exposed sensor area.

Both sensors were installed outdoors for two weeks and measurements were conducted. The sensors were previously deposited with NaCl using a spray nozzle. The amounts of NaCl deposited were 0.05 and 0.5 g/m^2^. The sensors were installed in a transparent acrylic stand with good ventilation, so that the salt was not washed away by rain and so that the sensors were affected by solar radiation.

Observations were conducted in the Abiko area of Chiba, Japan (35.9° N, 140.0° E). The distance from the coastline was sufficiently far (25 km) that the amount of airborne sea salt was considered negligible. A dummy plate was placed in the observation platform to assess the effect of airborne salinity. After the observation, the dummy plate was wiped, the deposits were dissolved in water, and the electrical conductivity was measured. The results showed that the NaCl equivalent was 0.014 g/m^2^, which means that the amount was small compared to the 0.5 g/m^2^ deposited. Observations were carried out for two weeks from 23 July 2024.

Electrical resistance was measured using the four-terminal method with an RM3545-02 (HIOKI). A weather meter was also installed next to the observation platform to measure temperature, humidity, and solar radiation. In addition, a thermocouple was installed on the product sensor, as shown in Figure 2, to determine the temperature rise on the sensor surface due to solar radiation.

## 3. Results

### 3.1. Comparison of Sensors

Figure 3 shows the corrosion depths measured by the product sensors over two weeks, as well as the respective variations in air temperature, sensor surface temperature, relative humidity, and solar radiation in the first four rows. The temperature of the sensor part was more likely to be higher because the reference part was covered by a coating. During the day, the surface temperature of the sensor was almost 20 °C higher than the air temperature. At this time, the temperature of the reference part was not as high because of the covering. This resulted in a relatively low electrical resistance of the sensor section and a spike error in the corrosion loss during the day. This means that product sensors can only be used in environments without daylight or in the long term.

The corrosion loss when corrected by the surface temperature of the sensor as measured by the thermocouples installed in the sensor part, rather than by the ratio of the sensor section to the reference section, is shown in the fifth row of Figure 3. The resistance value was corrected using the following formula, converted to the equivalent of 25 °C, and the corrosion loss was calculated from the ratio of the increase in resistance from the initial state:R_25_ = (1 + α∆T)R,(2)
where R_25_ is the equivalent resistance of a metal film at 25 °C, R is the measured resistance, ΔT is the difference between 25 °C and the measured temperature, and α is the temperature coefficient of iron.

Although the shape and magnitude of the error changed, it cannot be said that data with sufficient accuracy were obtained: the sensor part of the RCM sensor is a metal layer with a small thermal capacity, so the temperature fluctuates easily and the difference between the thermal response of the thermocouple and the sensor still remains as an error.

The corrosion loss data acquired by the improved sensor are then shown in the sixth column of Figure 3. The spike-like errors during the day have disappeared, and the errors caused by temperature differences have been significantly reduced.

If the daytime error part of the product sensor is excluded, the results are in good agreement in the long term, indicating that the corrosion depth can also be measured with the improved sensor.

The detailed data focusing on one day for both sensors are shown in Figure 4 (specifically, the data are from 1 August, the 12th day of observation). From around 5:00 to 7:00, the corrosion depth measured by the product sensor gradually increases in relation to that measured by the improved sensor. After a large and obvious error, the data of the product sensor gradually converge to match the improved sensor data from around 15:00 to 18:00. Because corrosion is a phenomenon in which the corrosion depth increases monotonically, this gradual divergence is an error of the product sensor.

Although there were some differences in the absolute corrosion depth for the 0.5 g/m^2^ trial, the long-term trends of the two sensors were nearly identical, as can be seen in Figure 3. This difference in absolute amount is thought to be caused by the time difference between the spraying of the salt water and start of measurement, as well as the initial oxide film.

The results show that the improved sensor can be used to measure corrosion depth with a higher accuracy even during the daytime, when it was not possible to do so before.

### 3.2. Calculation of Corrosion Rate

Temperature and humidity change from moment to moment in an outdoor environment. We thus extract the corrosion rate every hour with the improved sensor, as one hour is too short a time scale for corrosion processes and the mass losses are very small, which makes them very difficult to measure. If measurements can be performed with a time resolution of about one hour, the corrosion rate can be compared with the daily environmental changes.

As discussed in Section 3.1, the daytime errors have been significantly improved, but errors remain, which make it difficult to extract the corrosion rate. Figure 5 shows the measured data for the corrosion depth, with a more detailed expansion of Figure 4. The data are plotted for 1 August, with a salt amount of 0.5 g/m^2^ and improved sensor conditions as an example. Corrosion rate extraction is possible for periods of small temperature fluctuations during the night, but the corrosion rate cannot be obtained in this way during the day. The following process was therefore used to calculate the hourly corrosion rates.

First, as shown by the black dotted line in the figure, the data are divided into hourly intervals, and a linear approximation is made for each period. The variance is then calculated for the difference between the original data (blue) and the approximation (black), and intervals with variance above a given threshold are determined to be error periods. Figure 6 shows histograms of the differences between the approximation and the original data from 4:00 to 5:00 and 7:00 to 8:00 as examples.

Here, the threshold of variance is set to be 10^−6^. For the periods judged not to be low error, the gradient of the approximation line is taken as the hourly corrosion rate. The data for the periods determined to be high error are then linearly approximated from the endpoints of low error periods. The results of this process are plotted as red lines in Figure 5. In periods with a high measurement accuracy, the approximate line is almost identical to the original data, while in periods with large errors, the corrosion rate is interpolated by linear approximation into a form from which it can be extracted. This enables the hourly corrosion rate to be obtained even in sections with large errors during the daytime.

The upper panel of Figure 7 shows the measured and fitted data for the corrosion depth over the whole period for both salt amounts. The data processing works well for all of the time intervals in the experiment.

The middle panel of Figure 7 shows the hourly corrosion rates obtained by using the method described above. Appropriate corrosion rates were obtained for the 0.5 g/m^2^ salt amount trials. However, on the other hand, the measurement accuracy was insufficient in the 0.05 g/m^2^ salt amount trials, as the corrosion rate was very slow and was comparable in magnitude to the error due to the temperature difference, resulting in a negative corrosion rate. For current sensors, it is difficult to measure the corrosion rate every hour with a daily average of less than 50 µm/year. However, it is possible that the design of the adhesion between the sensor and the reference part could be improved to enable such measurements.

The relative humidity of the air and the relative humidity of the sensor surface are shown in the lower panel of Figure 7. Relative humidity is an important parameter because electrolytes form on metal surfaces because of the deliquescence of deposited salts. The relative humidity of the sensor surface is the parameter that effectively affects corrosion, as the temperature of the metal surface differs significantly from the air temperature because of solar radiation [20]. The relative humidity of the sensor surface was calculated from the air temperature, the relative humidity of the air, and the sensor surface temperature using the following equations:(3)$${RH}_{surf}={RH}_{air}\frac{P_{H_{2}O,sat}\left(T_{air}\right)}{P_{H_{2}O,sat}\left(T_{surf}\right)}$$
(4)$$P_{H_{2}O,sat}\left(T\right)=0.61121\exp\left(\left(18.678-\frac{T}{234.5}\right)\left(\frac{T}{257.14+T}\right)\right)$$

Here, $T_{air}$ is the temperature of the air in °C, ${RH}_{air}$ is the relative humidity of the air in kPa, $T_{surf}$ is the temperature of the sensor surface, ${RH}_{surf}$ is the relative humidity of the sensor surface, and $P_{H_{2}O,sat}$ is the saturation pressure of water vapor.

Over a daily cycle, the corrosion rate is very low during the day and progresses at night. This is due to the electrolyte drying out under the influence of solar radiation. The corrosion per day also decreases with time. This is due to the rust layer formed on the metal surface, which reduces the corrosion rate. There is a clear correlation between the relative humidity on the sensor surface and the corrosion rate. In particular, during the night of 29 July, the corrosion rate temporarily decreased in response to a temporary decrease in relative humidity.

## 4. Conclusions

To reduce errors in ER sensors caused by temperature differences between the sensor part and the reference part, we constructed an improved sensor where the sensor part overlapped with the reference part. Comparative tests between the improved sensor and existing products were carried out, and errors in the form of intense spikes during the daytime were successfully reduced. Furthermore, by using a data-filtering process, we were able to measure the corrosion rate every hour. Hourly corrosion rate measurements were difficult when the average daily corrosion rate was less than 50 µm/year under conditions of 0.05 g/m^2^ salt.

As described above, the improved sensor was developed to compare high-temporal-resolution corrosion rates with the environmental changes from moment to moment. If the atmospheric corrosion rate, which until now could only be evaluated in the long term, can be evaluated on the same time scale as changes in temperature and relative humidity, a detailed identification of the environmental variables with a significant influence on the corrosion process will be made possible.

As a next step, this error can be further reduced by designing a thinner adhesive part between the sensor and reference parts. The adhesive used in this study was 150 µm double-sided tape, but if this were reduced to 10 µm, the time constant for the reduction in the temperature difference between the two metals could be made smaller by a factor of approximately 1/225 (or 1/15^2^), assuming the same heat transfer coefficient for the adhesive. This would also allow for even lower corrosion rates to be assessed under solar radiation.

## Figures and Tables

**Figure 1 sensors-25-00268-f001:**
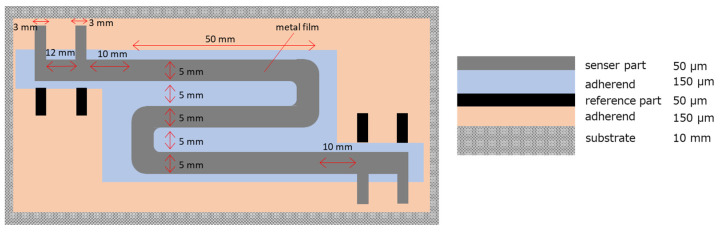
Design of improved ER sensor. Right is the top view and left is the side view.

**Figure 2 sensors-25-00268-f002:**
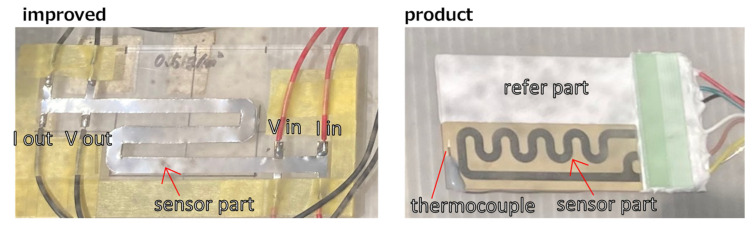
Picture of the improved sensor (**left**) and existing product (**right**). Thermocouples were attached to the surface of the existing product.

**Figure 3 sensors-25-00268-f003:**
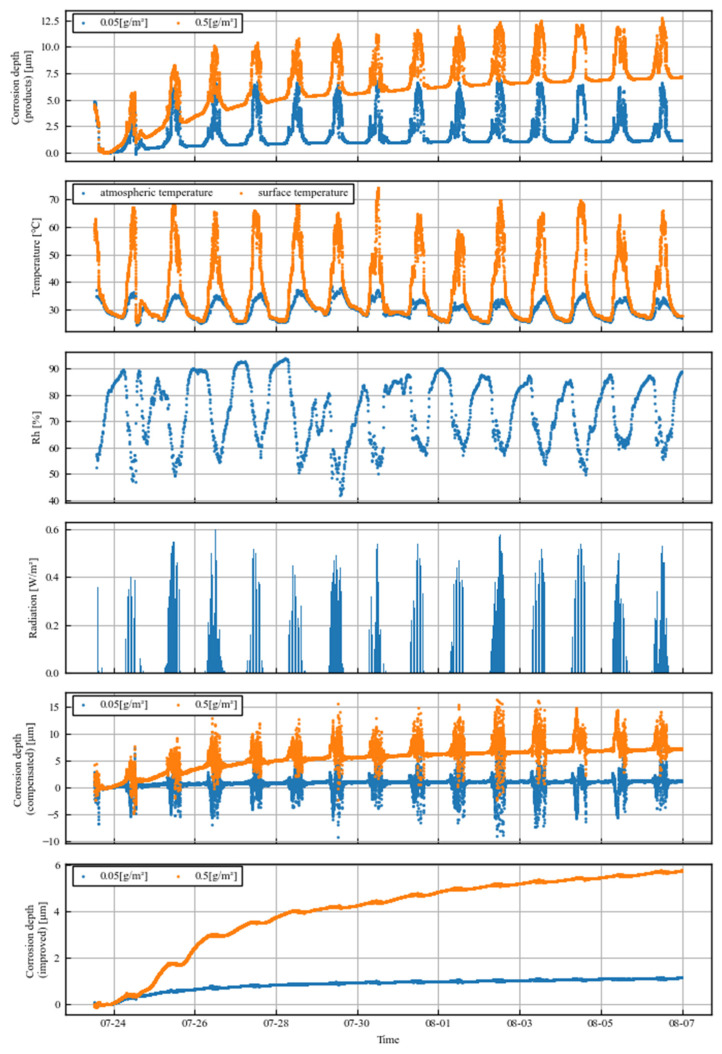
Time series data of observations. From top to bottom: corrosion depth measured with existing products, temperature, humidity, solar radiation, corrosion depth compensated for with surface temperature, and corrosion depth measured with improved sensors.

**Figure 4 sensors-25-00268-f004:**
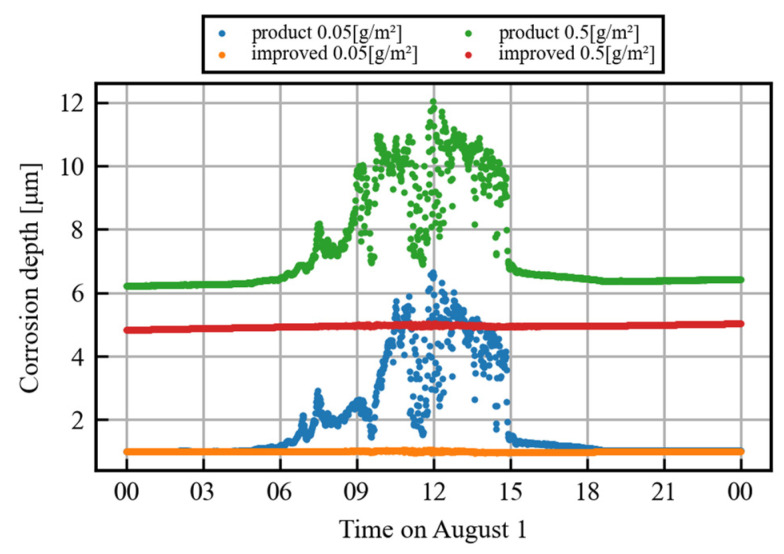
Comparison of measured daily corrosion depths between product and improved sensors.

**Figure 5 sensors-25-00268-f005:**
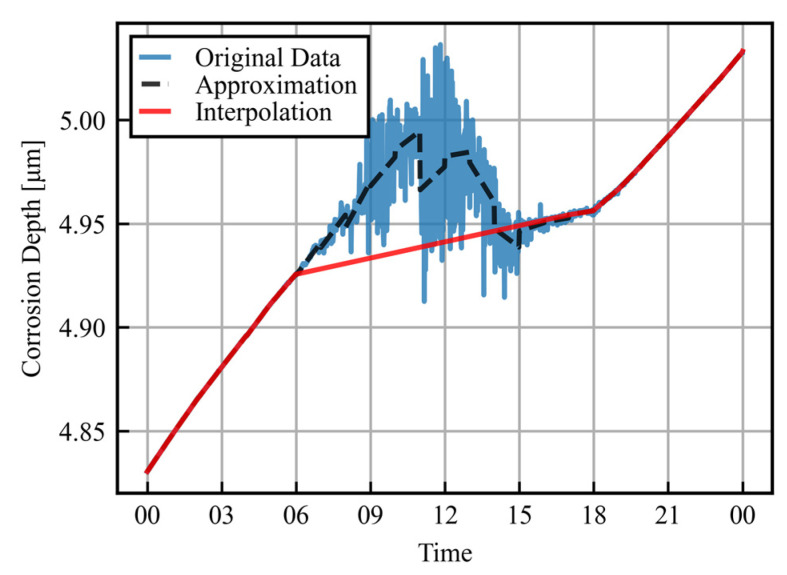
Detailed corrosion depth for one day: raw data and linear approximation after data processing.

**Figure 6 sensors-25-00268-f006:**
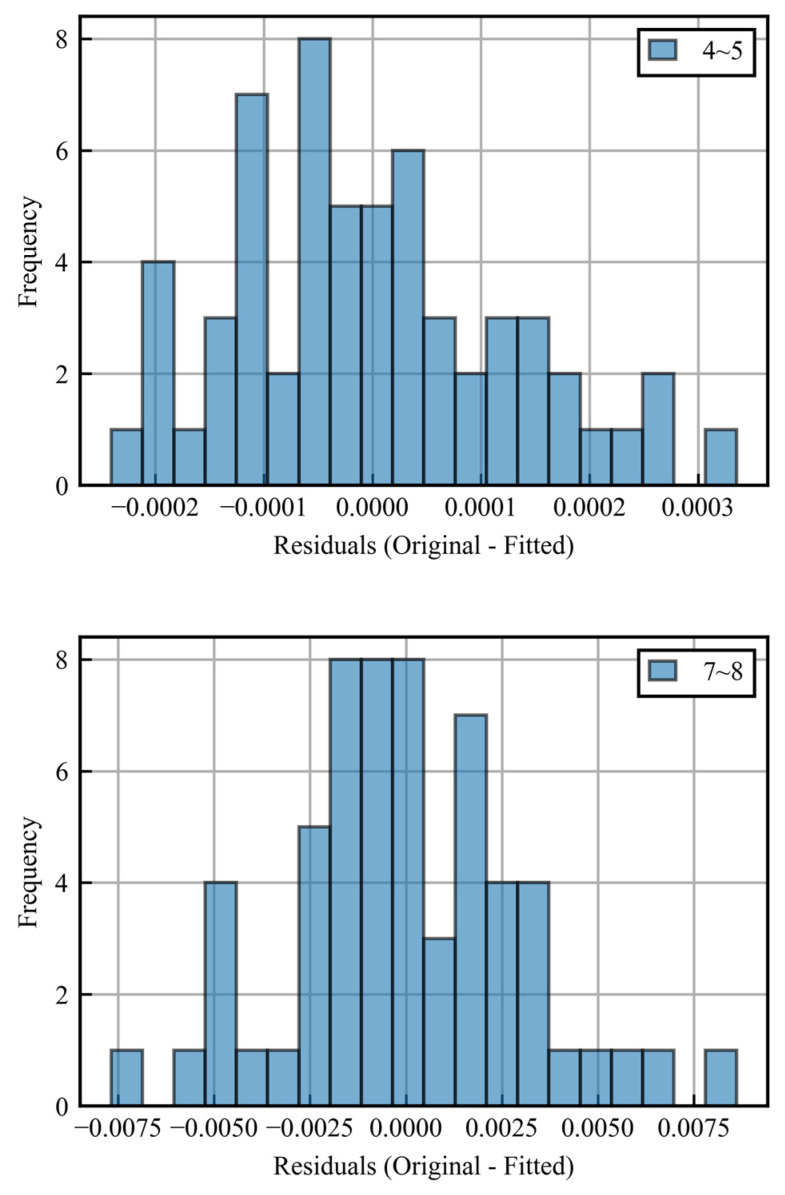
Histograms of residuals for correct period and error period.

**Figure 7 sensors-25-00268-f007:**
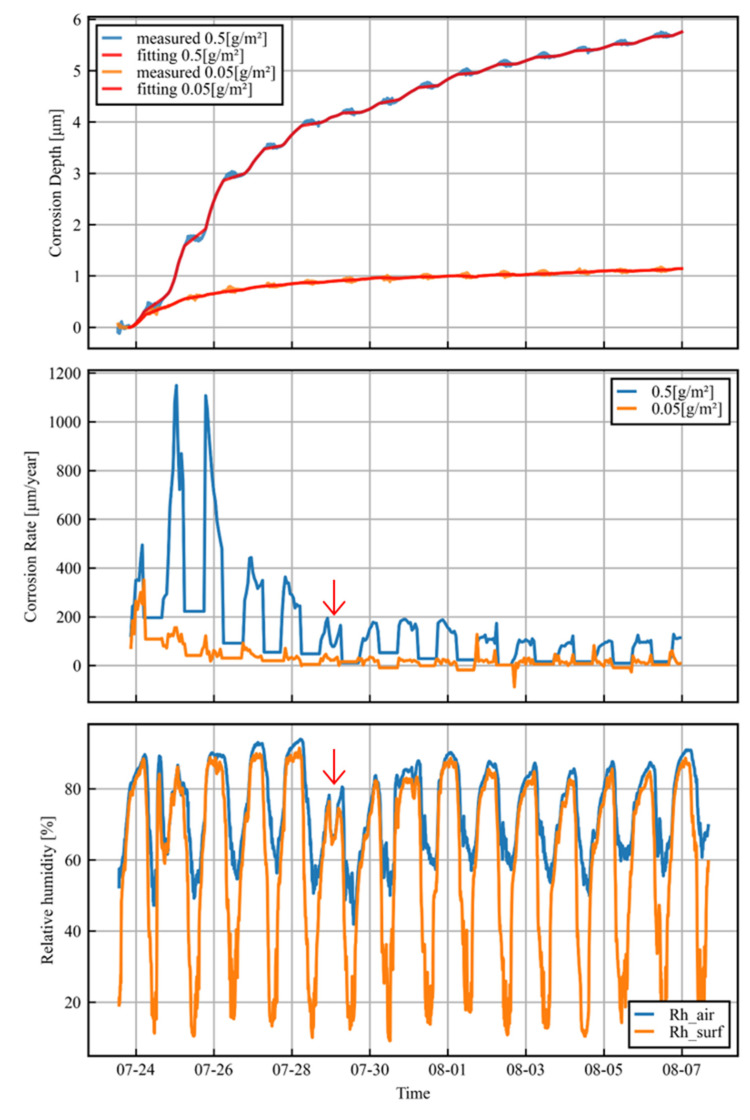
Time series of corrosion depth, corrosion rate, and relative humidity.

## Data Availability

The author confirms that the data supporting the findings of this study are available within the article.

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
