# Peer review of "High-Temporal-Resolution Corrosion Monitoring in Fluctuating-Temperature Environments with an Improved Electrical Resistance Sensor"

_sensors, 2025, doi:10.3390/s25010268_

Round 1
Reviewer 1 Report
Comments and Suggestions for Authors
In the reviewed paper, the author focused on the experimental investigation of a high temporal resolution of the improved ER sensor in fluctuating temperature environments. The paper is clearly structured, well-discussed and sound.
The main issue of the paper is that it does not have a minimum recommended length for the Article (16 pages) but just for communication. In terms of content, this manuscript also belongs to the classification of communication paper.
Another problem with the text is that the list of references needs to be expanded.
The third major shortcoming is that the sensor testing site (Abiko area of ​​Chiba, Japan) is not linked to the ISO 9223-2012 standard. Even If the previous classification data (C1-C5, CX was excluded in the text) did not exist, then the discussion and conclusion should have made a connection between the measured data and the classification. Even without the previous one, the conclusion should have been minimally linked (lines 258-260) “were able to measure the corrosion rate every hour. Hourly corrosion rate measurements were difficult when the average daily corrosion rate was less than 50 μm/year under conditions of 0.05 g/m2 salt.” in that sense (under conditions of 0.05 g/m salt, the sensor was not applicable for categories C1, C2 and C3 but only C4 (high corrosivity) and C5 (very high corrosivity) according to the standard mentioned above (ISO 9223-2012, Corrosion of metals and alloys - Corrosivity of atmospheres - Classification, determination and estimation).
English is generally good but needs to be polished.
A list of all comments is given below. Some of them are corrections, and some are just suggestions.
Line 10: measure the corrosion rate accurately instead of accurately measure corrosion rate
Line 16: measured instead of were able to measure
Line 20: environmental changes instead of changes in the environment
Line 37: Add a comma before “such as cable [8]”
Line 71: “to measure the corrosion depth accurately with high temporal resolution.” instead of “to accurately measure the corrosion depth with high temporal resolution.”
Line 85: Consider “impossible” instead of “not possible”.
Line 86: Consider “occasionally” instead of “from time to time”.
Line 104: 1) No type* of the tape is declared.
Although not obligatory, it could be stated.
*chemical composition
2) Thickness looks high. Is there any reason for this (relatively) tick tape? Even for the insulation purposes 2 mils (50 micrometers) should be good enough.
Finally, the conclusion led to the same (even stronger) point.
Line 105: It is not obligatory, but the purity of iron is a useful detail. Is it standard 99.5 (99.8)% Fe purity or very high purity (99.995%)? If you do not have an analysis, the specification would be enough.
Line 121: Here, the period of the year the experiment was carried out should be stated, or after the first sentence of the previous paragraph (... two weeks and measurements were conducted.)
Readers will see this in the images below, but it must also be stated in the experimental section.
Line 131: Add commas before and after “as shown in Figure 2”
Line 139: Please add a space between the numeric value and its unit.
Do it for every temperature, e.g.., lines 151, 152, etc.
Line 164 (Figure 3): X-axes should start from 07-24 (with the space between 07-23 and y-axes, it would be okay not to have 07-23 as the date). Additionally, the ticks at X-axes should be two days every time, not just one, as in the case of 07-31 and 08-01.
Line 169: Delete a space (after from and before a comma); Please state the date of the 12th day, it is hard to observe it from Figure 3.
Lines 173-174: Very nice observation (conclusion). Not everyone gives this kind of data, which should be the case more frequently. No need to be corrected.
Line 178: Just a suggestion. Consider “nearly identical” instead of “almost the same”.
Line 189: daily environmental changes instead of environmental changes each day.
Line 194: Add “with” before “a salt amount”.
Line 280 (References): Although the Introduction section is well written, the author needs to add more references. Even for a short article (communication?), about 20 references or more would still be required.
The comments are also given in the attached pdf document.

Reviewer 2 Report
Comments and Suggestions for Authors
This study improved version of the electrical resistance (ER) sensor that can acquire corrosion rates even under temperature fluctuations by eliminating the temperature difference between the reference part and the sensor part was developed. To decrease the error, authors developed an improved sensor composed of a reference metal film and an overlaid sensor metal film to cancel temperature differences between them. The improved sensor was compared with an existing sensor product in outdoor monitoring experiments. The authors need to incorporate the following few modification/suggestions in the manuscript also clarify following few queries.
1. The author needs to supplement some references to more clearly and logically explain the advantages and disadvantages of ER sensors in measuring atmospheric corrosion rates.
2. The author needs to provide more detailed explanations of the differences in design and manufacturing of the improved ER sensor, and mark them on the diagram to show.
3. Improved ER sensor utilize sensor part and reference part implementation testing.So authors should give the specific values of the resistance values for the sensor part and the reference part as a function of temperature and humidity.
4. There are some errors in the Manuscript that need to correct. For example, Figure 1“sense part” Spelling mistakesï¼›chapter 3.1 “Figure 2” should be changed to “Figure 3”.
5. The author should provide the corrosion depth of the real sample in the corrosive environment as a reference
6. To 0.5 g/m2 trial there is a difference in the measurement results of the improved sensor and the product at night, but 0.05 g/m2 trial no similar phenomenon The article argues that “This difference in absolute amount is thought to be caused by by the time difference between the spraying of the salt water and start of measurement, as well as the initial oxide film.” The authors should check the experimental procedures to ensure that the initial states of the two sensors are consistent.
7. Figure 6 chooses to provide the error histogram for periods 7-8, from Figure 5, the error for periods 7-8 is not large in the error period, is it representative? Authors should also provide error histograms for periods 11-12.
8. The author should explain the rationality of the calculation method from Original Data to Approximation to Interpolation. The difference between the initial data and the interpolation corrosion depth of the 6-15 in Figure 5 is large.
Comments on the Quality of English Languageaverage
Round 2
Reviewer 2 Report
Comments and Suggestions for Authors
No more comments